# *Macrocystis pyrifera* Extract Residual as Nutrient Source for the Production of Sophorolipids Compounds by Marine Yeast *Rhodotorula rubra*

**DOI:** 10.3390/molecules26082355

**Published:** 2021-04-18

**Authors:** Allison Leyton, Michael Araya, Fadia Tala, Liset Flores, María Elena Lienqueo, Carolina Shene

**Affiliations:** 1Center for Biotechnology and Bioengineering (CeBiB), Center of Food Biotechnology and Bioseparations, BIOREN and Department of Chemical Engineering, Universidad de La Frontera, Francisco Salazar 01145, Temuco 4780000, Chile; liset.flores@ufrontera.cl (L.F.); Carolina.shene@ufrontera.cl (C.S.); 2Centro de Investigación y Desarrollo Tecnológico de Algas y otros Recursos Biológicos (CIDTA), Facultad de Ciencias Marinas, Universidad Católica del Norte, Coquimbo 17811421, Chile; mmaraya@ucn.cl (M.A.); ftala@ucn.cl (F.T.); 3Departamento de Biología Marina, Universidad Católica del Norte, Larrondo 1281, Coquimbo 17811421, Chile; 4Coastal Socio-Ecological Millenium Institute (SECOS), Santiago 8370459, Chile; 5Center for Biotechnology and Bioengineering (CeBiB), Department of Chemical Engineering, Biotechnology and Materials, Universidad de Chile, Beauchef 851, Santiago 8370459, Chile; mlienque@ing.uchile.cl

**Keywords:** brown seaweed, sophorolipid, antibacterial activity, *Macrocystis pyrifera*, marine microorganism, *Rhodotorula mucilaginosa*

## Abstract

Seaweed processing generates liquid fraction residual that could be used as a low-cost nutrient source for microbial production of metabolites. The *Rhodotorula* strain is able to produce antimicrobial compounds known as sophorolipids. Our aim was to evaluate sophorolipid production, with antibacterial activity, by marine *Rhodotorula rubra* using liquid fraction residual (LFR) from the brown seaweed *Macrocystis pyrifera* as the nutrient source. LFR having a composition of 32% *w*/*w* carbohydrate, 1% *w*/*w* lipids, 15% *w*/*w* protein and 52% *w*/*w* ash. The best culture condition for sophorolipid production was LFR 40% *v*/*v*, without yeast extract, artificial seawater 80% *v*/*v* at 15 °C by 3 growth days, with the antibacterial activity of 24.4 ± 3.1 % on *Escherichia coli* and 21.1 ± 3.8 % on *Staphylococcus aureus.* It was possible to identify mono-acetylated acidic and methyl ester acidic sophorolipid. These compounds possess potential as pathogen controllers for application in the food industry.

## 1. Introduction

In their natural environment, microorganisms have to compete for space and nutrients; thus, the capability to produce antimicrobial compounds is a trait developed by selection pressure. In yeast strains, the production of sophorolipid compounds with antibacterial activity has been reported. Sophorolipids (SLs) are extracellular amphipathic glycolipids that consist of a disaccharide, sophorose (2-*O*-β-d-glucopyranosyl-β-d-glucopyranose), and a hydroxyl fatty acid [1,2]. These compounds act on the cell membrane, causing destabilization and increasing permeability [3]. SLs can have the presence of lactonic or acidic forms; the lactonic form of the carboxyl group of the fatty acid moiety is esterified to the sophorose, and the acidic form of the carboxyl group of the fatty acid moiety is not esterified [4]. Also, the fatty acid chain present in SLs might vary in size (mostly between C16 and C18) with the presence of unsaturation (saturated, monounsaturated, or polyunsaturated) [5]. This mix of structurally related molecules in SLs influence their properties and applications. Lactonic forms are more hydrophobic [6] and have been reported to have better biocide activities [7], anticancer [8], spermicide, cytotoxic, and proinflammatory activities [9]. On the other hand, acidic forms are better foaming agents, have higher water solubility [10], and have been reported to present better use in the food industry, bioremediation, and cosmetics [11].

The production of SLs is strongly stimulated when two sources of carbon (hydrophobic and hydrophilic) are present in the medium [4]. In the absence of a hydrophobic carbon source, low production levels had been observed [12]. The expression of these compounds can be greatly influenced by environmental manipulations as by the culture medium composition [13,14,15]. SLs are produced by several yeast species such as *Torulopsis bombicola* (formerly *Candida bombicola*), *Starmerella apicola* (formerly *Candida apicola*), *Pseudohyphozyma bogoriensis* (formerly *Rhodotorula bogoriensis*), and *Wickerhamiella domercqiae* [2].

*Pseudohyphozyma (Rhodotorula*) is a yeast aerobic capable of synthesizing numerous metabolites useful in industries, such as lipids, carotenoids, and sophorolipids. Species, such as *P. bogoriensis* and *R. babjevae*, are able to produce sophorolipid compounds with antibacterial and antifungal activity [14,16,17]. Its clear advantage is the capacity to grow and synthesize metabolites on substrates from industrial raw material wastes, which considerably elevates the economic profitability of biotechnological processes. In previous work, it was possible to isolate a marine *Rhodotorula* strain from brown seaweed tissue, which has the capacity of using the brown seaweed biomass as a nutrient source for growth and metabolite production [18].

The brown seaweed *Macrocystis pyrifera* (giant kelp) is an important economic resource from natural populations used as raw materials for the alginate industry [19]. In order to improve the sustainability in the exploitation of *M. pyrifera,* successful cultivation at pilot scales has been achieved in Chile [19,20,21]. *M. pyrifera* has been used as a chemical platform for the production of varied biocompounds via conversion of the carbohydrate fraction through microbial fermentation [22,23,24,25] or directly used in pharmaceuticals and food applications [26]. An example is the phlorotannin extraction processes from *M. pyrifera* that produce a liquid fraction residual (LFR), which was used as a microbial nutrient source to produce a carotenoid compound [18,27].

In the present work, we have evaluated the sophorolipid compounds production with antibacterial activity by the marine yeast *Rhodotorula rubra* (formerly *Rhodotorula mucilaginosa*) using LFR from *Macrocystis pyrifera* as a nutrient source. Additionally, the effect of culture conditions on the production of biomass and antibacterial compounds by marine *R. rubra* was performed.

## 2. Results

### 2.1. Characterization of LFR

The composition of LFR was 32% *w*/*w* of carbohydrate (hydrophilic sources of carbon), 1% *w*/*w* of lipids (hydrophobic sources of carbon), 15% *w*/*w* of protein, and 52% *w*/*w* of ash (Table 1). The ash present in the extract could be salts and minerals of the biomass algal, mostly, and/or formed during the extraction process (alkaline extraction and pH adjustment). The ash fraction did not need to be removed prior to its use as a nutrient source.

### 2.2. Effect of Culture Conditions on the Production of Biomass and Antibacterial Compounds by the Marine R. Rubra

The effect of the incubation time and temperature and the composition of the growth medium (artificial seawater, ASW) on the production of extracellular antibacterial sophorolipid compounds by *R. rubra* was evaluated. In the first set of experiments, the antibacterial activity was evaluated only on *E. coli* (Figure 1). The incubation temperature (15 and 25 °C) had no significant (*p* > 0.05) effect on the production of biomass and antibacterial compounds. Due to the similar temperature present in the natural environment where yeast was isolated, the low temperature (15 °C) was selected for yeast growth. The best incubation time to produce antibacterial compounds was 3 d (exponential growth phase), while the biomass concentration was higher after the incubation was carried out for 6 d. Because our objective was to improve the production of antibacterial compounds, the following experiments were conducted using a 3 d incubation period. Finally, for the higher production of antibacterial compounds, the ASW concentration was 80% *v*/*v*.

In the second set of experiments, the concentration of LFR and yeast extract on the production of antibacterial compounds was evaluated (Table 2).

In these experiments, the antibacterial activity was tested on two pathogenic bacteria, *E. coli* and *S. aureus* (Figure 2 and Figure 3). The highest antimicrobial activity was obtained when *R. rubra* grew in LFR 40% *v*/*v,* without yeast extract, with the antibacterial activity of 24.4 ± 2.1 % inhibition on *E. coli* and 21.1 ± 2.8 % inhibition on *S. aureus* (Figure 2b, run 6). The lowest antibacterial activity was obtained when *R. rubra* was grown in LFR 40% *v*/*v,* yeast extract 6 g/L (Figure 2b, run 5), with the antibacterial activity against *E. coli* and *S. aureus* of 19.5 ± 2.1 and 10.5 ± 2.1 % inhibition, respectively. This low antibacterial activity could be related to the increase in the biomass concentration, 4.9 ± 0.1 g/L (the highest, Figure 2a), in comparison to the concentration obtained under conditions with high antimicrobial activity, 2.9 ± 0.3 g/L (Figure 2a, run 6). The same tendency for the consumption of carbohydrates in LFW was presented, 0.71 ± 0.12 g/L vs 0.51 ± 0.03 g/L (Figure 2a).

The lowest biomass concentration was obtained when *R. rubra* grew in LFR 30% *v*/*v*, without yeast extract (Figure 2a, run 4); conditions related with lowest LFR consumption and the absence of yeast extract in the culture medium. The lowest LFR consumption suggests a possible growth inhibition.

Figure 3 shows the results of the antibacterial activity of sophorolipid compounds against *E. coli* and *S. aureus.* The SLs inhibited the growth of both pathogen bacteria, resulting in a clear zone (“halo”) around the discs (Figure 3, run 5 and run 6 Table 2). In contrast, the LFR did not exhibit inhibitory activity or negative control (Figure 3, C sector).

### 2.3. Identification of Sophorolipid with Antibacterial Activity Produced by R. Rubra

The chromatographic analysis identifies two SLs in major proportions from the extracellular extract of *R. rubra*—mono-acetylated (1AC) acidic sophorolipid (AS) AS C22:0-1AC and the methyl ester acidic sophorolipid AS C22:1 (Figure 4). Figure 4a shows the mass spectrum in full scan mode for AS C22:0-1AC. The *m*/*z* corresponding to the ammonium, sodium, and potassium adducts were [M + NH4] + *m*/*z*: 740.4677, [M + Na] + *m*/*z*: 745.4236, and [M + K] + *m*/*z*: 761.3900, respectively. Figure 4b shows the confirmatory mass spectrum for the fragmentation of [M + NH4] + *m*/*z*: 740.4656; therefore, the mass [M + H] + *m*/*z*: 723.4394 is observed, which corresponds to the mass of the phospholipid plus one in positive mode. The mass spectrum of Figure 4b is essential because it indicates that the mass 740.4656 corresponds to the sophorolipid with an ammonium adduct; otherwise, the mass 723.43 and its fragments would not be observed, and other masses would be seen instead.

Figure 4c shows the mass spectrum in full scan mode AS C22:1. The *m*/*z* corresponding to the ammonium, sodium, and potassium adducts were [M + NH4] + *m*/*z*: 696.4381, [M + Na] + *m*/*z*: 701.3940, and [M + K] + *m*/*z*: 717.3681, respectively. Figure 4d shows the confirmatory mass spectrum for the fragmentation of [M + NH4] + *m*/*z*: 696.4381; therefore, the mass [M + H] + *m*/*z*: 679.4111 is observed, which corresponds to the mass of the phospholipid plus one in positive mode.

## 3. Discussion

This study reports the antibacterial sophorolipids producing ability of a hitherto unreported marine yeast, *R. rubra,* using liquid fraction waste from brown seaweed *M. pyrifera*. The nutrient source (LFR) employed for the growth of *R. rubra* is a complex mix of nutrients with two sources of carbon (hydrophobic and hydrophilic), 1% *w*/*w* of lipids, and 32% *w*/*w* of carbohydrates, respectively, which would favor the SLs production by the marine yeast. In their study [28], de Oliveira et al. showed that a low N/C ratio (< 0.5) enhanced the production of SLs, while that biomass production was stimulated when the N/C ratio was near to 1 [29,30]. Our results show that the extract with the highest antibacterial activity was obtained when the marine yeast, *R. rubra,* was cultivated with only LFR in a high concentration (40% *v*/*v*) without yeast extract; the condition of a low N/C ratio in comparison to the culture medium supplemented with an extra nitrogen source (yeast extract).

Low-cost nutrient sources to produce SLs in the *Candida* strain included sugar cane molasses [31], soy molasses [32], and deproteinized whey [33], obtaining an SLs concentration of 23, 21, and 422 g/L, respectively. In this case, due to the marine origin of the yeast *R. rubra*, isolated from the *M. pyrifera* tissue [18], the use of LFR as a by-product derived to phlorotannin extraction from brown seaweed *M. pyrifera* is presented as a good alternative to produce sophorolipid compounds. The use of LFR as a nutrient source allowed us to obtain a low concentration of SLs, 0.1 g/L (Table 2), compared to the *Candida* strain. This difference could be due to the use of a culture medium supplemented with oil, in addition to the use of fed-batch culture mode in the *Candida* strain; therefore, there is hope that the use of this culture condition allows the increase of SLs production by *R. rubra* (work in progress).

SLs production also depends on the culture conditions [15]. In our study, the highest antibacterial SLs production was observed at 72 h, during the exponential phase, indicating a growth-associated antibacterial production. Similar growth-associated production of SLs was also obtained by Sen et al. [17] when they cultivated *Rhodotorula babjevae* YS3 in a medium containing glucose at 19 °C; a similar temperature using in this study (15 °C). Daniel et al. [33] had reported that the production of SLs with *Starmerella apicola,* using hydrolyzed broth from the biomass of the yeast *Cutaneotrichosporon curvatum (formerly Cryptococcus curvatus)* as culture medium, began in the middle of the exponential growth phase.

The production of SLs for the *Rhodotorula* strain has been reported by limited. Nuñez et al. [8] and Ribeiro et al. [14] reported the production of 22 carbon SLs in *R. bogoriensis*, with concentrations between 0.5–1 g/L. In *R. rubra,* the production of SLs has been reported by Chandran and Das [34] but without indicating the concentration produced. In the marine yeast strain, the SLs production has not been described. On the other hand, when comparing the production of SLs by genus, *Candida* vs *Rhodotorula*, the production of the longer carbon chain of SLs compound in the *Rhodotorula* strain was observed (C16–C18 in *Candida* vs C22–C24 in *Rhodotorula*), which is related to a greater antimicrobial activity [16,34].

The fatty acid profile in brown seaweed is rich in unsaturated fatty acids, such as docosenoic acid (C22:1) and saturated fatty acid such as docosanoic acid (C22:0) [35,36]. Both fatty acids could have been directed to incorporate into the structure of SLs synthetized by the marine *R. rubra* [1], and at the same time, the methyl esterification of the fatty acid fraction could promote the production of SLs mainly in the acidic form (AS) in comparison with the lactonic form (LS) [37]. The production of sophorolipid compounds of C22:0 and C22:1 carbon in the *Rhodotorula* strain had been previously reported [14,15,16].

The mass spectra analysis shows that the common structure of SLs produced by the marine *R. rubra* is the 13-[2-O-β-D-glucopyranosyl- β-D-glucopyranosyloxy]-docosanoic acid SLs; this structure has already been described for the *Rhodotorula* strain [14,16].

Finally, the antibacterial activity against *Staphylococcus aureus* and *Escherichia coli* indicates a potential application as a pathogen controller in the food industry. A similar activity was also observed by Dengle-Pulate et al. [38] against *Staphylococcus aureus* and *Bacillus subtilis* during their study involving SLs produced by *Starmerella apicola* using glucose as the hydrophilic source and lauryl alcohol C12–14 as the hydrophobic source.

## 4. Materials and Methods

### 4.1. Liquid Waste Fraction (LWR) from M. Pyrifera

Fronds of *Macrocystis pyrifera*, including stipe, gas bladder, and blades, were collected by scuba diving 30 km Southwest from Puerto Montt, Chile. The samples harvested were dried at 40 °C and ground to an average size lower than 1.4 mm. The LWR was obtained from the phlorotannins extraction process, according to Leyton et al. [25], as is described in Figure 5. The phlorotannins extraction was made with NaOH 0.5 mol/L, using a seaweed mass-to-liquid ratio of 1/20 weight/volume, *w*/*v*, (180 min, 100 °C). The suspension was centrifuged (2057× *g*, 20 min) (Centrifuge 5804R Eppendorf AG, Hamburg, Germany), and the pH of the liquid phase was adjusted to 7.0 with HCl. Phlorotannins, which could act as cell growth inhibitors, were removed by adsorption on a macroporous resin (Amberlite XAD-16N, Sigma-Aldrich, St. Louis, MO, USA). The extract (200 mL) was incubated with the resin (40 g) under agitation (150 rpm, 25 °C) (orbital shaker Zhicheng, model ZHWY-211B, Shanghai, China) for 12 h [32]. The resin was removed by filtration and the liquid phase, LWR, was kept at −20 °C until use.

### 4.2. Microorganism and Inoculum Preparation

The *Rhodotorula rubra* strain (GenBank accession no KU167831) was isolated and identified in previous work [18]. The strain was kept at −20 °C in glycerol 50% *v*/*v* stock until use.

One mL of the yeast stock culture was seeded in 50 mL of a basal sterile medium of the following composition: yeast extract (Bacto TM) 2 g/L and glucose (Merck) 10 g/L in 60% *v*/*v* artificial seawater (ASW) [39]. The incubation was made for 3 d at 25 °C under continuous orbital shaking at 150 rpm. Then, 1 mL of the grown culture was used to inoculate a new flask with 50 mL of seaweed extract-based medium of the following composition: LFW 40% *v*/*v*, ASW 60% *v*/*v,* and yeast extract 2 g/L. The incubation was made at 25 °C, 150 rpm, for 3 d. All the culture experiments used this inoculum.

### 4.3. Effect of Culture Conditions on the Production of Antibacterial Compounds

The effect of incubation time (3 and 6 d), temperature (15 and 25 °C), and ASW (20%, 60%, and 80% *v*/*v*) on the antibacterial activity of the *R. rubra* culture extract was determined. The effect of each of the factors was evaluated changing one factor at a time, keeping the rest factors at the level specified for the basal conditions (incubation time 6 d, incubation temperature 25 °C, yeast extract concentration 6 g/L, ASW concentration 60% *v*/*v,* and LWR at 40% *v*/*v*).

The effect of LFR (20%, 30%, and 40 % *v*/*v*) and yeast extract (0 and 6 g/L) concentration was evaluated in the second set of experiments in which the incubation time and temperature, and the concentration of ASW defined in the first set of experiments were used.

In each set of experiments, two Erlenmeyer flasks were prepared with 50 mL of the sterile-specified medium and each flask was inoculated with 5 mL of the above-specified inoculum. After the incubation at the specified conditions, the total content of the flask was centrifuged (7000× *g*, 4 °C, 10 min) to recover the biomass; the supernatant was stored at −20 °C until use.

### 4.4. Separation of Antibacterial Compounds

The cell-free culture broth (50 mL) was extracted with 20 mL of ethyl acetate under agitation (150 rpm, 15 °C, 1 h). The mixture was transferred to decanting funnel to allow phases of separation at room temperature for 1 h. The organic phase was vacuum-dried at 40 °C to remove the solvent. The dry sample was dissolved in 0.5 mL of methanol and stored at −20 °C until antimicrobial assay.

### 4.5. Evaluation of the Antimicrobial Activity

The antimicrobial activity of the extracellular extract (EE) of *R. rubra* was evaluated by the disk diffusion method against one Gram-positive bacteria (*Staphylococcus aureus* ATCC25923) and one Gram-negative bacteria (*Escherichia coli* ATCC25922) in Mueller–Hinton agar medium (Sigma Aldrich). Pathogen assays were prepared by diluting the 24 h culture grown in 10 mL of sterile Luria–Bertani (LB) broth to cell density diluted of 10^6^ UFC/mL.

Twenty µL of the bacterial sample was deposited into sterile paper discs (5 mm in diameter of Whatmann N°1) and placed on a Petri dish containing the pathogen in Mueller–Hinton agar medium. Positive control was ampicillin (100 mg/L), and negative control was LWF. Plates were kept to 4 °C for 1 h to allow the diffusion of chemicals; after, these plates were incubated at 37 °C for 12 h. Inhibition zones were measured using software ImageJ (ImageJ bundled with 64-bit Java 1.8.0_112) and expressed as a percent of inhibition, antibacterial activity.

### 4.6. Assay Methods

Characterization of LWR. The characterization of LWR was estimated indirectly from the chemical composition of the initial biomass less. The chemical composition of residual biomass was obtained later in the phlorotannins extraction process. The moisture content, as well as protein, lipid, ash, and fiber contents in the seaweed biomass were quantified following the official methods of the Association of Official Analytical Chemistry (AOAC): 930.04, 978.04, 991.36, 930.05, and 962.09, respectively [40]. The non-nitrogen extract was estimated as the difference between 100 and the sum of the percentages of protein, ash, lipid, and fiber content.

*Biomass concentration*. The biomass concentration of *R. rubra* was determined gravimetrically; a known culture volume (2 mL) was centrifuged (2057× *g*, 10 min; CT15E, Himac, Hitachi, Tokyo, Japan), and the cell pellet was washed twice with distilled water, centrifuged, and dried at 105 °C until constant weight.

*Determination of the total carbohydrate concentration.* Total carbohydrate concentration was measured using the phenol-sulfuric acid method [41]. Briefly, 200 µL of the sample was added to a 200 µL phenol solution (5 *w*/*v* %) and supplemented with 1 mL of concentrated sulfuric acid. The mixture was equilibrated for 20 min at room temperature. The absorbance was measured at 476 nm against a water blank. A calibration curve of glucose at different concentrations (0.02–0.1 g/L) was prepared; the results were expressed as grams of glucose per liter of LFR. The consumption of LFR by the difference between the initial and final concentration was determined.

*Identification of sophorolipids compounds.* UHPLC system (Dionex UltiMate 3000) coupled to mass spectrometry (Orbitrap^®®^ Q Exactive Focus detector from Thermo Scientific was used. A Gemini^®®^ 3 µm NX-C18 column (50 × 2 mm) and a gradient elution at 0.35 mL/min were used for the separation. The two solvents used were HPLC water (phase A) and acetonitrile (phase B); both phases contained 6.7 mM of ammonia. Conditions of the elution gradient were: 0%–10% B 1.8min, 30% B 3min, 90% B 6 min, 90% B 7.5 min; 15% B was where it remained constant for 1.5 min. Finally, phase B dropped to 15%, where it remained constant for 4.5 min. Detection was carried out in electrospray positive ion mode (ESI+). Measurements were recorded in full scan mode, scan range 150–1000 *m*/*z,* and microscan tree scan per second and then confirmed by MS/MS. The performance of fragmentation was at CE 20 eV. The ESI condition was sheath gas flow rate 35, aux gas flow rate 10, sweep gas flow rate 0, spray voltage 3.5 kV, capillary temperature 350 °C, S-lens RF level 100, and aux gas heater temperature 250 °C.

### 4.7. Statistical Analysis

All the culture experiments were made in duplicate. The results are presented as average ± standard deviation. The significance and relative influence of each individual factor on the production of *R. rubra* biomass and antibacterial compounds were determined using the variance analysis (ANOVA). The significance of the factors was determined at 5% confidence level.

## 5. Conclusions

Sophorolipids production with antibacterial activity on *E. coli* and *S. aureus* from marine *R. rubra* that was cultivated in a growth medium based on liquid fractions residual (LFR) from the brown seaweed *Macrocystis pyrifera* was demonstrated. Sophorolipid compounds were identified as a mono-acetylated acidic (AS C22:0-1Ac) and a methyl ester acidic (AS C22:1) sophorolipid. The synthesis of sophorolipid compounds by the marine *R. rubra* depends on the culture medium composition and especially on the type of carbohydrate source present.

## Figures and Tables

**Figure 1 molecules-26-02355-f001:**
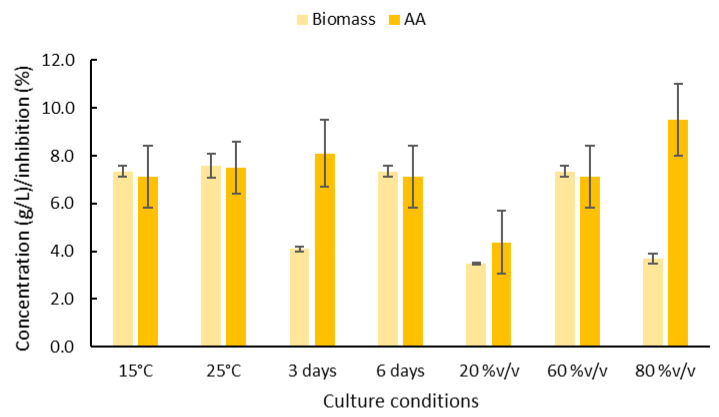
Effect of culture conditions: incubation temperature (15 and 25 °C), incubation time (3 and 6 days), and growth medium composition: artificial seawater concentration (20, 60, and 80 %*v*/*v*) on the biomass concentration and antibacterial activity on *E. coli* (AA) of the extract from cultures of *R. rubra*.

**Figure 2 molecules-26-02355-f002:**
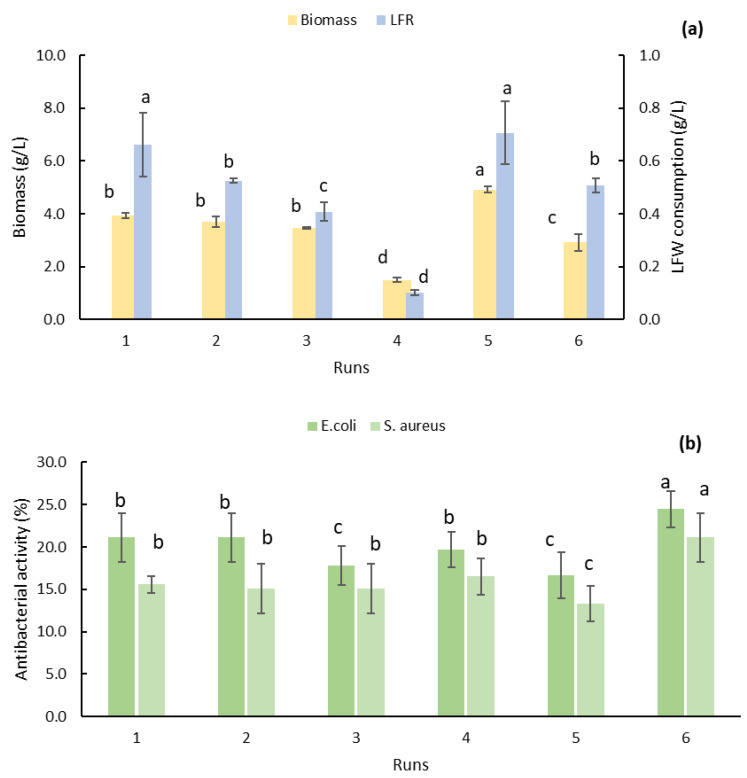
Effect of LFR and yeast extract concentration on biomass concentration and LFR consumption (**a**) and antibacterial activity (**b**) in cultures of *R. rubra*. Symbols indicate statistical differences (*p* < 0.05).

**Figure 3 molecules-26-02355-f003:**
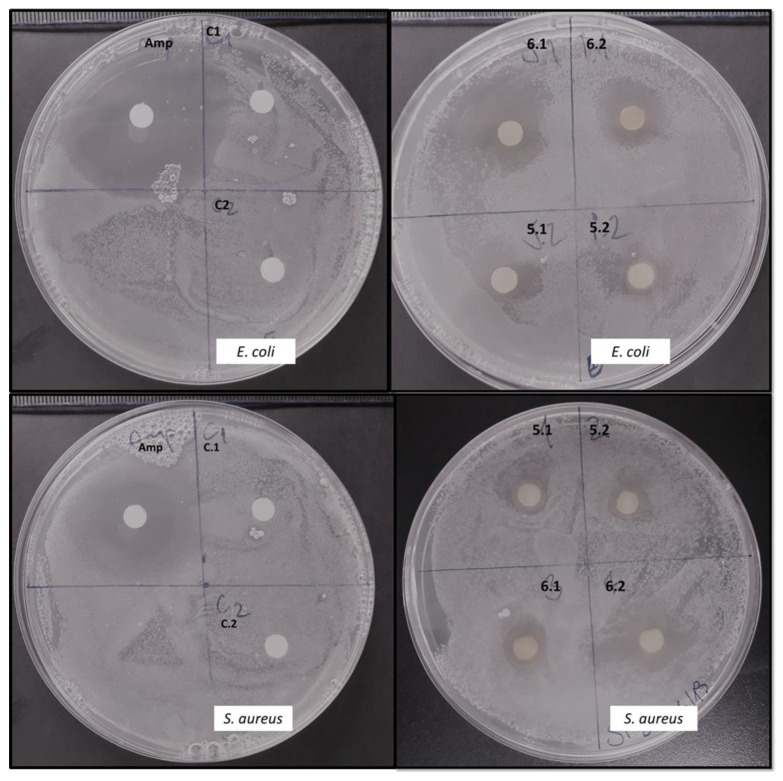
Antibacterial activity of the extract from cultures of *R. rubra* against *Staphylococcus aureus* ATCC25923 and *Escherichia coli* ATCC25922 in Mueller–Hinton agar medium. Amp: ampicillin (positive control); C: LFR (negative control); run 6 and run 5 extract from cultures of *R. rubra*, 1 and 2 are duplicates of the samples.

**Figure 4 molecules-26-02355-f004:**
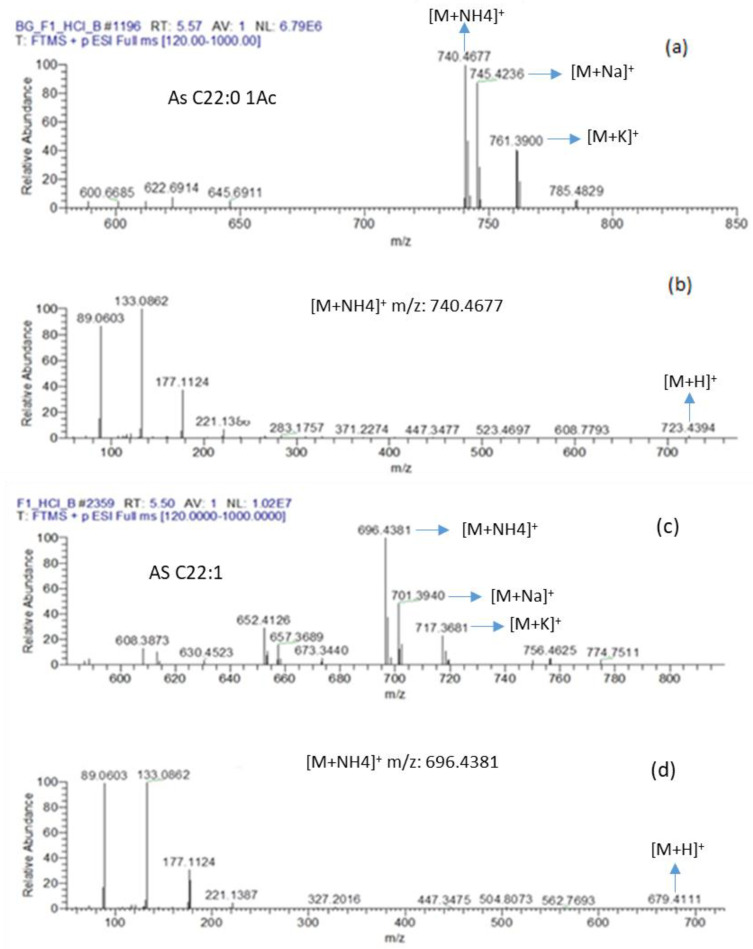
Full scan LC-MS mass spectrum of the *R. rubra* extract with antibacterial activity under culture conditions LFR 40% *v*/*v*, without yeast extract, seawater 80% *v*/*v*, incubation temperature of 15 °C and incubation time of 3 days. (**a**) *m*/*z* for AS C22:0-1AC; (**b**) confirmatory mass spectrum for the fragmentation of [M + NH4] + *m*/*z*: 740.4656; (**c**) *m*/*z* for AS C22:1 and (**d**) confirmatory mass spectrum for the fragmentation of [M + NH4] + *m*/*z*: 696.4403. AS: acidic sophorolipid; AC: acetyl group.

**Figure 5 molecules-26-02355-f005:**
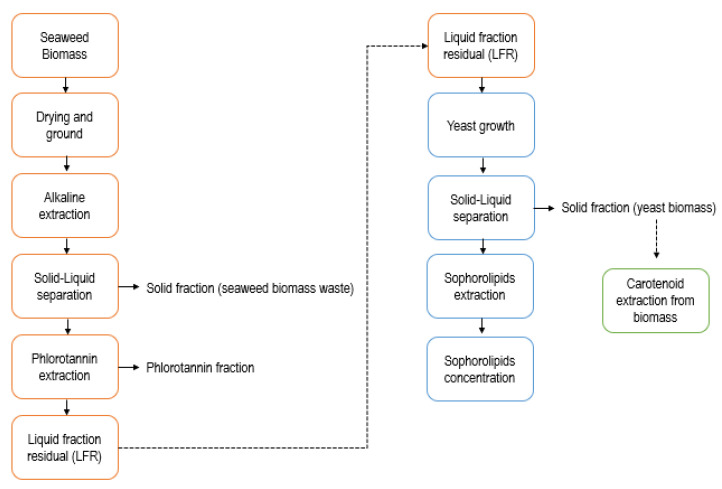
Liquid waste fraction (LFR) production and its sequential use for the production of antimicrobial compounds by the marine *Rhodotorula rubra.*

**Table 1 molecules-26-02355-t001:** Chemical composition of LFR from *M. pyrifera*.

Chemical Composition	Unit	Seaweed Extract
		Dry Basis
Lipids	%	1.4 ± 0.1
Ash	%	51.5 ± 1.5
Protein	%	15 ± 0.5
Carbohydrates*	%	32 ± 1.0

Carbohydrate* estimated from no nitrogen extract + fiber.

**Table 2 molecules-26-02355-t002:** LFR and yeast extract concentration evaluated on biomass and antibacterial sophorolipids production by marine *R. rubra*.

Runs	LFR (%)	YE (g/L)	SLs (g/L)
1	20	6	0.12
2	20	0	0.13
3	30	6	0.09
4	30	0	0.12
5	40	6	0.07
6	40	0	0.20

## Data Availability

Not applicable.

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
