# Peer review of "Macrocystis pyrifera Extract Residual as Nutrient Source for the Production of Sophorolipids Compounds by Marine Yeast Rhodotorula rubra"

_molecules, 2021, doi:10.3390/molecules26082355_

Round 1

Reviewer 1 Report

molecules-1183104-rev

The manuscript entitled "Macrocystis pyrifera extract waste as nutrient source for the production of sophorolipids compounds by marine yeast Rhodotorula mucilaginosa" addresses a relevant topic on the production of antimicrobial compounds from yeasts living in the brown algae fronds Macrocystis pyrifera. This manuscript is well structured, well written, although there is a need to correct several errors at the level of taxonomy, which I indicate below.

Corrections needed:

Title - Macrocystis pyrifera extract waste as nutrient source for the production of sophorolipids compounds by marine yeast Rhodotorula rubra

line 4 - Rhodotorula rubra

line 17 - Rhodotorula rubra

line 33 - sophorose (2-O-β-D-glucopyranosyl-β-D-glucopyranose), 

line 40/41 - Torulopsis bombicola (formerly Candida bombicola)

line 41 - Starmerella apicola (formerly Candida apicola), Pseudohyphozyma bogoriensis (formerly Rhodotorula bogoriensis) and Wickerhamiella domercqiae

line 42 - Pseudohyphozyma (and Rhodotorula) is a yeast aerobic capable ...

line 43 - P. bogoriensis

line 61 - Rhodotorula rubra (formerly Rhodotorula mucilaginosa)

line 75 - R. rubra (in italics)

line 78 - Rhodotorula rubra

line 81 - R. rubra

line 95 - R. rubra

line 100 - R. rubra

line 104 - R. rubra

line 106 - R. rubra

line 116 - R. rubra

line 123 - R. rubra

line 125 - R. rubra

line 129 - R. rubra

line 140 - R. rubra

line 147 - R. rubra

line 148 - R. rubra

line 154 - R. rubra

line 160 - R. rubra

line 168 - Starmerella apicola

line 169 - Cutaneotrichosporon curvatum (formerly Cryptococcus curvatus)

line 187 - R. rubra

line 192 - Starmerella apicola

line 210 - Rhodotorula rubra

line 222 - R. rubra

line 242 - R. rubra

line 296 - R. rubra

line 300 - R. rubra

Author Response

The manuscript entitled "Macrocystis pyrifera extract waste as nutrient source for the production of sophorolipids compounds by marine yeast Rhodotorula mucilaginosa" addresses a relevant topic on the production of antimicrobial compounds from yeasts living in the brown algae fronds Macrocystis pyrifera. This manuscript is well structured, well written, although there is a need to correct several errors at the level of taxonomy, which I indicate below.

Thanks, by your comments

Corrections needed:

Title - Macrocystis pyrifera extract waste as nutrient source for the production of sophorolipids compounds by marine yeast Rhodotorula rubra

Change performed in the title. Line 4

line 4 - Rhodotorula rubra

Change performed in line 4

line 17 - Rhodotorula rubra

Change performed in line 17

line 33 - sophorose (2-O-β-D-glucopyranosyl-β-D-glucopyranose), 

Change performed in line 33

line 40/41 - Torulopsis bombicola (formerly Candida bombicola)

Change performed in line 50

line 41 - Starmerella apicola (formerly Candida apicola), Pseudohyphozyma bogoriensis (formerly Rhodotorula bogoriensis) and Wickerhamiella domercqiae

Change performed in line 50 to 51

line 42 - Pseudohyphozyma (and Rhodotorula) is a yeast aerobic capable ...

Change performed in line 53

line 43 - P. bogoriensis

Change performed in line 55

line 61 - Rhodotorula rubra (formerly Rhodotorula mucilaginosa)

Change performed in line 72

line 75 - R. rubra (in italics)

Change performed in line 75

line 78 - Rhodotorula rubra

Change performed in line 91

line 81 - R. rubra

Change performed in line 105

line 95 - R. rubra

Change performed in line 109

line 100 - R. rubra

Change performed in line 113

line 104 - R. rubra

Change performed in line 115

line 106 - R. rubra

Change performed in line 128

line 116 - R. rubra

Change performed in line 135

line 123 - R. rubra

Change performed in line 137

line 125 - R. rubra

Change performed in line 139

line 129 - R. rubra

Change performed in line 141

line 140 - R. rubra

Change performed in line 158

line 147 - R. rubra

Change performed in line 166

line 148 - R. rubra

Change performed in line 167

line 154 - R. rubra

Change performed in line 173

line 160 - R. rubra

Change performed in line 179

line 168 - Starmerella apicola

Change performed in line 191

line 169 - Cutaneotrichosporon curvatum (formerly Cryptococcus curvatus)

Change performed in line 192-193

line 187 - R. rubra

Change performed in line 196

line 192 - Starmerella apicola

Change performed in line 215

line 210 - Rhodotorula rubra

Change performed in line 237

line 222 - R. rubra

Change performed in line 252

line 242 - R. rubra

Change performed in line 272

line 296 - R. rubra

Change performed in line 294

line 300 - R. rubra

Change performed in line 327

Reviewer 2 Report

In the present study, the authors evaluated sophorolipid production, with antibacterial activity, by marine Rhodotorula mucilaginosa using liquid fraction waste (LFW) from the brown seaweed Macrocystis pyrifera as a nutrient source. The authors found that the best culture condition for sophorolipid production was LFW 40% v/v, without yeast extract, artificial seawater 80% v/v at 15°C by 3 growth days, with antibacterial activity of 24.4 ± 3.1 % on Escherichia coli and 21.1 ± 3.8 21 % on Staphylococcus aureus. The sophorolipid was possible to be identified as mono-acetylated acidic and methyl ester acidic sophorolipid by LC-ESI/MS/MS. This manuscript is well written and provides important findings. However, there are some points needed to be addressed before publication.

1.In the introduction, sophorolipid is the main compound of this manuscript. However, the description of sophorolipid regarding its structure, production, biological functions, and commercial applications seems to be insufficient.

2.Line 49, use? or using?

3.Can the authors describe what is the usage of Rhodotorula mucilaginosa biomass (yeast biomass in Figure 1)?

4.Need the ash in LFW be removed? Why?

5.Lines 68 and 69, 32% w/w of carbohydrate (hydrophilic carbohydrate), 1% w/w of lipids (hydrophobic carbohydrate), I consider this sentence needed to be revised since the difference between 32% and 1% is very huge. Moreover, lipids are not carbohydrate, this may mislead the readers.

6.In Figure 2, the antibacterial activity measured is for E. coli, this point should be added to the legend of Figure 2.

7.In Figure 3(a), the authors need to discuss why the biomass and LFW consumption in run 4 is the lowest.

8.In the legends of Figures 2, 3, the description of statistics needs to be added.

9.Line 120, FLW? or LFW?

10.In the legend of Figure 4, “6 and 5”, “1 and 2”, revise them to “run 6 and run 5”, “run 1 and run 2”, is better.

11.Can the production yield of sophorolipid be calculated and presented?

12.I am not clear about Figure 5, is Figure 5 (a) R. mucilaginosa extract or AS C22:0-1AC? what peak in Figure 5 (a) represents AS C22:0-1AC? In Figure 5 (b), why this confirmatory mass spectrum is necessary? There are same questions in Figure 5 (c) and Figure 5 (d).

13.Lines 256-258, this sentence is not clear. The authors need to rewrite it.

Author Response

In the present study, the authors evaluated sophorolipid production, with antibacterial activity, by marine Rhodotorula mucilaginosa using liquid fraction waste (LFW) from the brown seaweed Macrocystis pyrifera as a nutrient source. The authors found that the best culture condition for sophorolipid production was LFW 40% v/v, without yeast extract, artificial seawater 80% v/v at 15°C by 3 growth days, with antibacterial activity of 24.4 ± 3.1 % on Escherichia coli and 21.1 ± 3.8 21 % on Staphylococcus aureus. The sophorolipid was possible to be identified as mono-acetylated acidic and methyl ester acidic sophorolipid by LC-ESI/MS/MS. This manuscript is well written and provides important findings. However, there are some points needed to be addressed before publication.

Thanks, by your comments. the points indicated were improved or changed

1.In the introduction, sophorolipid is the main compound of this manuscript. However, the description of sophorolipid regarding its structure, production, biological functions, and commercial applications seems to be insufficient.

It was added more information about sophorolipids. Please see line 35 to 44.

2.Line 49, use? or using?

Word changed by “using”

3.Can the authors describe what is the usage of Rhodotorula mucilaginosa biomass (yeast biomass in Figure 1)?

It was added this information. Please see figure 1 in chapter 4.1

4.Need the ash in LFW be removed? Why?

Added line 82-83: The ash fraction need not removed previous to the use as nutrient source.

5.Lines 68 and 69, 32% w/w of carbohydrate (hydrophilic carbohydrate), 1% w/w of lipids (hydrophobic carbohydrate), I consider this sentence needed to be revised since the difference between 32% and 1% is very huge. Moreover, lipids are not carbohydrate, this may mislead the readers.

Word carbohydrate change by carbon source, see line 78-79

6.In Figure 2, the antibacterial activity measured is for E. coli, this point should be added to the legend of Figure 2.

It was added in the legend of Figure 2 (now figure 1).

7.In Figure 3(a), the authors need to discuss why the biomass and LFW consumption in run 4 is the lowest.

Line 122-125: The lowest biomass concentration was obtained when R. rubra grow in LFR 30% v/v and without yeast extract (Figure 2a, run 4), conditions related with lowest LFR consumption and the absence of yeast extract in the culture medium. The lowest LFR consumption suggesting a possible growth inhibition.

8.In the legends of Figures 2, 3, the description of statistics needs to be added.

Figure 2 now is 1, and Figure 3 now is 2. The description was added line 128-129.

9.Line 120, FLW? or LFW?

FLW was change by LFR, liquid fraction residual

10.In the legend of Figure 4, “6 and 5”, “1 and 2”, revise them to “run 6 and run 5”, “run 1 and run 2”, is better.

In the legend was added word “Run”, line 135-138

11.Can the production yield of sophorolipid be calculated and presented?

The yield was presented in the table 2.

12.I am not clear about Figure 5, is Figure 5 (a) R. mucilaginosa extract or AS C22:0-1AC? what peak in Figure 5 (a) represents AS C22:0-1AC? In Figure 5 (b), why this confirmatory mass spectrum is necessary? There are same questions in Figure 5 (c) and Figure 5 (d).

The injections were made directly with the extracts. The figure 5 now is Figure 4. The figure 4 and the paragraph were improved.

Line 142-146 “The masses of figure 5a, correspond to the form of the sophorolipid As C22: 0 1Ac. The m/z corresponding to the ammonium, sodium and potassium adducts are [M + NH4] + m / z: 740.4677, [M + Na] + m / z: 745.4236, [M + K] + m / z: 761.3900 respectively.Figure 5b shows the confirmatory mass spectrum for the fragmentation of [M + NH4] + m/z: 740.4656, therefore the mass [M + H] + m / z: 723.4394 is observed, which corresponds to the mass of the phospholipid plus one in positive mode. The mass spectrum of figure 5b is essential, because it indicates that the mass 740.4656 corresponds to the sophorolipid with ammonium adduct, otherwise the mass 723.43 and its fragments would not be observed, other masses would be seen”

Similar description is for Figure 4c and 4d.

13.Lines 256-258, this sentence is not clear. The authors need to rewrite it.

Line 282-285 “The characterization of LWR was estimated indirectly from chemical composition of initial biomass less chemical composition of residual biomass obtained later of the phlorotannin extraction process. The moisture content, as well as protein, lipid, ash and fiber contents in the seaweed biomass were quantified”

Reviewer 3 Report

Why the authors prepared the Macrocystis pyrifera extract in the laboratory and did not use industrial waste? In this case, the use of the term "waste" throughout the article is wrong.

Figure 1 should be moved to Chapter 4.1.

The data in Figure 2 should be presented in one cumulative column graph using, for example, color differentiation.

‘Pathogen assays were prepared by diluting the 24 h culture grown…’ and figure 4 - the growth of bacteria on the plates is not uniform, as can be seen in the photographs. The agar medium was not thoroughly mixed before it solidified. This undermines the readings and their reliability. Then the zones of growth inhibition are not well visible. If performed correctly, the experiments should have the shape of a circle, which confirms that the experiment was performed incorrectly.

Too few bacteria have been tested to determine antimicrobial activity.

Author Response

 Thanks, by your comments

Why the authors prepared the Macrocystis pyrifera extract in the laboratory and did not use industrial waste? In this case, the use of the term "waste" throughout the article is wrong.

Because the industrial phlorotannin extraction process (where is obtaining the liquid fraction waste), from brown seaweed, is not performed in the actually in Chile. The process phlorotannin extraction is propose in previous work. The liquid fraction is a residual fraction from phlorotannin extraction process. For more clarity changed waste by residual

Figure 1 should be moved to Chapter 4.1.

Figure 1 was moved to chapter 4.1

The data in Figure 2 should be presented in one cumulative column graph using, for example, color differentiation.

Figure 2 (now Figure 1) was change and improved.

‘Pathogen assays were prepared by diluting the 24 h culture grown…’ and figure 4 - the growth of bacteria on the plates is not uniform, as can be seen in the photographs. The agar medium was not thoroughly mixed before it solidified. This undermines the readings and their reliability. Then the zones of growth inhibition are not well visible. If performed correctly, the experiments should have the shape of a circle, which confirms that the experiment was performed incorrectly.

Figure 4 (now Figure 3), the plates in the Figure 3 were improved.

Too few bacteria have been tested to determine antimicrobial activity.

The research group only had E. coli and S. aureus as pathogen bacteria.

Round 2

Reviewer 2 Report

1.In Figure 2, the meaning of the symbols (*, **, ***, ****) and (+, ++, +++, ++++) should be addressed.

2.In Table 1, is there any standard deviation in these data?

Reviewer 3 Report

The currently valid name of the microorganism tested in this work is RHODOTORULA MUCILAGINOSA. Rhodotorula rubra is a synonym for this name, hence I do not understand why another reviewer suggested changing the correct name (currently applicable all over the world) to a synonym. This can be confusing as the use of synonyms is not recommended. For example, in The Yeasts: A Taxonomic Study (Kurtzman 2011) we can also find information that R. rubra is synonymous of R. babjevae. The nomenclature of the main microorganism in the manuscript should revert to its original form, R. mucilaginosa.